# Decreased Administration of Life-Sustaining Treatment just before Death among Older Inpatients in Japan: A Time-Trend Analysis from 2012 through 2014 Based on a Nationally Representative Sample

**DOI:** 10.3390/ijerph18063135

**Published:** 2021-03-18

**Authors:** Michi Sakai, Shosuke Ohtera, Tomohide Iwao, Yukiko Neff, Tomoe Uchida, Yoshimitsu Takahashi, Genta Kato, Tomohiro Kuroda, Shuzo Nishimura, Takeo Nakayama

**Affiliations:** 1Department of Health Informatics, Kyoto University School of Medicine and Public Health, Kyoto 606-8501, Japan; ootera.s.aa@niph.go.jp (S.O.); yukik24jp@yahoo.co.jp (Y.N.); uchida.tomoe.22s@st.kyoto-u.ac.jp (T.U.); takahashi.yoshimitsu.3m@kyoto-u.ac.jp (Y.T.); nakayama.takeo.4a@kyoto-u.ac.jp (T.N.); 2Division of Medical Information Technology and Administration Planning, Kyoto University Hospital, Kyoto 606-8507, Japan; tomohide@kuhp.kyoto-u.ac.jp (T.I.); tomo@kuhp.kyoto-u.ac.jp (T.K.); 3Department of Research, Institute for Health Economics and Policy, Tokyo 105-0003, Japan; shuzo@soleil.ocn.ne.jp; 4Solutions Center for Health Insurance Claims, Kyoto University Hospital, Kyoto 606-8507, Japan; qq9f8hn9@kuhp.kyoto-u.ac.jp

**Keywords:** aged, older adults, health insurance claims, end-of-life care

## Abstract

The administration of intensive end-of-life care just before death in older patients has become a major policy concern, as it increases medical costs; however, care intensity does not necessarily indicate quality. This study aimed to describe the temporal trends in the administration of life-sustaining treatments (LSTs) and intensive care unit (ICU) admissions just before death in older inpatients in Japan. We utilized the National Database of Health Insurance Claims and Specific Health Checkups of Japan (NDB). Inpatients who were aged ≥65 years and died in October of 2012, 2013, or 2014 were analyzed. The numbers of decedents in 2012, 2013, and 2014 were 3362, 3473, and 3516, respectively. The frequencies of receiving cardiopulmonary resuscitation (CPR) (11.0% to 8.3%), mechanical ventilation (MV) (13.1% to 9.8%), central venous catheter (CVC) insertion (10.6% to 7.8%), and ICU admission (9.1% to 7.8%), declined between 2012 and 2014. After adjusting for age, sex, and type of ward, the declining trends persisted for CPR, MV, and CVC insertion relative to the frequencies in 2012. Our results indicate that the administration of LST just before death in older inpatients in Japan decreased from 2012 to 2014.

## 1. Introduction

As populations around the world are rapidly ageing [1], the nature, quality, and costs of end-of-life care have attracted considerable interest [2,3,4,5,6,7,8]. The use of intensive end-of-life care just before the death of older patients has become a major policy concern, as it is considered a factor that increases medical costs [9]; however, care intensity does not necessarily indicate quality [10]. Since the late 1980s, ethical, legal, and social issues related to the withholding or withdrawal of intensive end-of-life care have been discussed in developed countries. In the 2000s, multiple policies and programs attempted to improve care at the end of life in the United States and European countries [11,12,13,14]. Japan has one of the world’s most elderly societies, with a high mortality rate [15]. In Japan, the proportion of people aged 65 years or older was 28.4%, and the mortality rate was 11.2 per 1000 population in 2019 [16]. Japan has also developed end-of-life care policies, such as guidelines on the decision-making process for care in the last stage of life [17] and hospice and palliative services [18]; other Asian counties, such as Korea [19] and Taiwan [20], have also developed end-of-life care policies. Research on temporal trends in the real-world practices regarding intensive care before death in older patients is now needed to examine the impact of these continued efforts in many countries.

A previous study in the U.S. targeting Medicare recipients who died between 1985 and 1999 showed that the proportion of decedents admitted to the ICU and treated with intensive procedures (e.g., feeding tube placements, intubations/tracheostomies, and receiving cardiopulmonary resuscitation increased [3]. However, between 2000 and 2015, there was an initial increase followed by a stabilization of the ICU admission rate [21]. Data from Washington State Death Certificates and the University of Washington medicine data warehouse, which includes clinical and administrative information from a university medical center, indicated that ICU admissions in the last 30 days of life significantly decreased from 2010 to 2015 among patients 65 years and older, suggesting the impact of the increased penetration of advance care planning and palliative care programs [22]. In the population-level cohort study of Canadian adults who died between 2004 and 2015, the proportion of patients admitted to the ICU in the last 6 months of life did not change, whereas the proportion receiving mechanical ventilation increased over the course of the 11 years [23]. A reliance on medical facilities to provide end-of-life care and a shortage of palliative, hospice or home-based care within the Canadian healthcare system were suggested as reasons for these trends [23]. However, less is known about the temporal trends in intensive care before death in the older Asian population.

Furthermore, limited evidence exists regarding the real-world practice with respect to intensive care “just before death” among older patients. According to a systematic review of studies of end-of-life resource utilization using health administrative data, the most common period defined as the end of life was the last 6 or 12 months [6]. As illness trajectories are quite variable over a long period of time before death [24], studies using longer periods may include treatments administered to patients who are not yet in the last stage of life. Thus, it is necessary to use a shorter period to elucidate the practices with regard to the provision of care just before death.

In this study, we aimed to describe the frequency of and temporal trends in the provision of intensive care just before death among older Japanese inpatients at the national level using the National Database of Health Insurance Claims and Specific Health Checkups of Japan (NDB), which contains almost 100% of the digitized health insurance claims [25].

## 2. Materials and Methods

### 2.1. Definitions

We defined the end-of-life as a specific period of time before death, which was recorded in health insurance claims data. Based on systematic reviews of studies that measured the intensity of end-of-life care [2,6], we defined intensive end-of-life care as receiving the following life-sustaining treatments (LSTs): cardiopulmonary resuscitation (CPR), mechanical ventilation (MV), and central venous catheter insertion (CVC) insertion or ICU admission.

### 2.2. Study Design and Data Source

We conducted a repeated cross-sectional study using data from the National Database of Health Insurance Claims and Specific Health Checkups of Japan (NDB) [25]. In Japan, all persons are required to be covered by public health insurance. As the NDB contains almost 100% of the digitized health insurance claims issued after the 2009 fiscal year, its use enables the evaluation of the real-world care practices in nearly the entire older Japanese population [26]. NDB data are untraceable and anonymized. The Ministry of Health, Labour and Welfare provides an NDB sampling dataset (NDB-SD) randomly sampled from 10% of the inpatient health insurance claims in the NDB for October, in which seasonal differences in healthcare utilization are not significant for research use. Because the date of administration is not recorded in claims issued before the 2012 fiscal year, we could not identify care practices in the last days of life before 2012. Thus, we used NDB-SD claims data issued from 2012 to 2014.

### 2.3. Subjects

The study subjects were deceased patients aged ≥65 years who had been admitted to a hospital in October of 2012, 2013, or 2014 for ≥7 days. As approximately 80% of care before death in Japan occurs in medical institutions, which is higher than in other countries [16], we focused on LSTs administered to hospitalized patients. Deceased patients were identified based on the hospitalization outcomes recorded in the claims, which allowed us to identify in-hospital mortality from claims data with a high degree of accuracy [27].

### 2.4. Response Variables

The response variables were the administration of CPR, the use of MV, CVC insertion and ICU admission in the last 7 days of life. We considered the administration of LSTs in the 7 days before death. Data on each LST were obtained from inpatient health insurance claims. CPR was either non-open cardiac massage, countershock, or life-saving endotracheal intubation. MV was either artificial respiration for life prolongation (invasive positive pressure ventilation, IPPV; noninvasive positive pressure ventilation, NPPV) or assisted ventilation for palliation, since we could not distinguish life-prolonging ventilation from other ventilation procedures due to the rules for reimbursement. ICU admissions were calculated from reimbursements for ICU admissions submitted by institutions providing acute care or emergency care.

### 2.5. Variables Used for Adjustment

We included age, sex, cancer diagnosis, and type of ward as factors for adjustment. Age was categorized into 65 to 74 years, 75 to 84 years, 85 to 94 years, and ≥95 years. We included cancer diagnoses because reimbursement incentives for palliative care applied to only cancer under the Japanese health insurance system could affect the practices regarding the administration of LSTs and ICU admissions. Cancer diagnoses were identified from the diagnoses recorded in the claims issued in the last month of life, i.e., October of each year. The types of wards in which patients died were categorized into general wards, long-term care wards, wards for psychiatric diseases, and wards for tuberculosis. General wards mainly provided acute care. Long-term care wards provided long-term care for patients with severe physical and cognitive problems.

### 2.6. Statistical Analysis

We applied the Cochran–Armitage trend test for detecting a temporal trend in the frequencies of LSTs and ICU admissions as we assume linear trend. The Cochran-Armitage (CA) test is used to test whether there is a linear trend when the response is binary. The null hypothesis is that the binomial proportion is the same for all levels of the explanatory variable. It is commonly used in epidemiology to test for temporal trend in healthcare utilizations. We used the administration of each LST and ICU admission as binary categorical response variables and the year of death as explanatory variable. We also conducted logistic regression analysis using the administration of each LST and ICU admission as response variables and the year of death, age, sex, cancer diagnosis, and type of ward as categorical variables for adjustment. As we assume that linear trend is different from 2012 to 2013 and 2013 to 2014, we treated the year of death as a categorical variable using 2012 as the reference year. We used SPSS version 23 (SPSS Inc., Chicago, IL, USA) for Windows for the analysis. We set the significance level of each test to 5%.

## 3. Results

### 3.1. Patient Characteristics

The numbers of hospitalized patients included in the NDB-SD for October 2012, October 2013, and October 2014 were 225,650, 225,709, and 227,970, respectively. The numbers of eligible patients for the analysis in each year were 3362, 3473, and 3516, respectively. We excluded patients aged <65 years (*n* = 2012: 83,380, 2013: 80,956, 2014: 79,423), patients who had an outcome other than death (i.e., continued, cure, termination, and transferred) (*n* = 2012: 136,852, 2013: 139,100, 2014: 142,861), patients with fewer than 7 days of hospitalization (*n* = 2012: 2054, 2013: 2178, 2014: 2168), and patients with no charge records for treatment (*n* = 2012: 2, 2013: 2, 2014: 2).

Comparisons of patient characteristics in each year are shown in Table 1. The proportions of male patients, the age distributions, and proportions of patients who received care in each ward were constant over the study period. Nearly half of the patients (44.2%, *n* = 1486 in 2012, 45.3%, *n* = 1575 in 2013, 48.2%, *n* = 1694 in 2014) died at 85 years of age or older.

### 3.2. Frequencies and Yearly Trends in the Administration of LSTs and ICU Admissions

The frequencies of the administration of CPR (11.0% to 8.3%; *p* for trend <0.001), the use of MV (13.1% to 9.8%; *p* for trend <0.001), CVC insertion (10.6% to 7.8%; *p* for trend <0.001), and ICU admission (9.1% to 7.8%; *p* for trend =0.29) declined from 2012 to 2014 (Table 2, Figure 1). Subgroup analyses stratified by age, sex, cancer diagnosis, and ward also indicated decreasing trends in almost all subgroups (excluding the psychiatric and tuberculosis wards, which had fewer patients). Larger declining trends in the administration of CPR and use of MV were identified for patients ≥95 years than for those aged between 65 and 74 years.

After adjusting for age, sex, cancer diagnosis, and ward type with logistic regression analysis, significant declining trends persisted for the administration of CPR (in 2013, adjusted odds ratio (AOR) 0.88 [95% CI 0.75–1.03]; in 2014, 0.74 [0.63–0.87]), the use of MV (in 2013, 0.89 [0.76–1.03]; in 2014, 0.74 [0.64–0.87]), and CVC insertion (in 2013, 0.82 [0.70–0.97]; in 2014, 0.75 [0.63–0.89]) relative to the values in 2012 (Table 3).

## 4. Discussion

We identified annual declining trends between 2012 and 2014 in the frequencies of the administration of CPR (11.0% to 8.3%, adjusted odds ratio in 2014 relative to 2012 (AOR) 0.74 [0.63–0.87]), the use of MV (13.1% to 9.8%, AOR 0.74 [0.64–0.87]), and CVC insertion (10.6% to 7.8%, AOR 0.75 [0.63–0.89]) just before death among patients aged ≥ 65 years using nationally representative claims data. Larger declining trends in the administration of CPR were observed in patients ≥ 95 years (7.9% to 4.0%) than in those aged between 65 and 74 years (9.8% to 9.7%).

Although the survey year, the period defined as the end of life, and study settings vary among studies, decreasing trends in intensive care are consistent with studies targeting decedents in Washington State in U.S. and in 14 European countries. Among older decedents in Washington State, ICU admissions in the last 30 days of life declined from 2010 to 2015 [22]. The increased penetration of advance care planning and palliative care since 2012 is suggested to have contributed to the decline [22]. Among patients who died in ICUs in 14 European countries, death without withholding LSTs significantly declined from 1999 to 2015 in accordance with the changes in European laws, recommendations and guidelines regarding end-of-life practices over the past decades [28]. Our results are also consistent with Japan’s rank in the Quality of Death Index published by The Economist Intelligence Unit, which evaluates developing and developed countries in terms of palliative and healthcare environment, human resources, affordability of care, quality of care, and community engagement [7,8]. Japan was ranked 23rd among 40 countries in 2010 [7] and 14th among 80 countries in 2015 [8]. Two Asian counties, Taiwan (at position 6 in 2015) and Singapore (at position 12) which developed a strong and effective national palliative care policy framework ranked higher than in Japan and also improved in rank in 2015 than in 2010 [8]. On the other hand, the frequencies of LSTs and ICU admission before the death of patients aged 65 years or older remain lower in Japan than those in countries that reported the care intensity at the national level. ICU admissions in the last 30 days of decedents in 2015 Medicare and decedents in 2009 Korean National Health Insurance Service were 27.4% [21], and 26.3% [19], respectively, whereas ICU admissions in the last 7 days in the present study were 7.8% in 2014. Administration of MV in the last 30 days of Korean decedents in 2009 was 19.5% [19], whereas MV in the last 7 days in the present study were 9.8% in 2014. Although the period defined as the end of life is longer than our study, differences in healthcare delivery systems may be also related to the discrepancy. Critical care services vary between countries in the number of beds. The number of ICU beds per 100,000 population in 2005 was 20.9 in the USA and 4.3 in Japan [29]. Korean healthcare delivery relies heavily on the private providers, leading to an increase in demands for intensive care [19].

The possible reason for the declines in the use of MV and CVC insertion in Japan is the improvements made to less invasive medical devices. Nasal high-flow therapy as an alternative to invasive positive pressure ventilation (IPPV) or noninvasive positive pressure ventilation (NPPV) and peripherally inserted central catheters (PICCs) as an alternative to CVCs might be related to these trends. However, the declining trend in the administration of CPR is worthy of attention, as improvements in medical devices are unlikely to be related to the frequency of the administration of CPR, which was identified based on reimbursement for either non-open cardiac massage, countershock, or life-saving endotracheal intubation. The larger declining trend in the administration of CPR in the group aged 95 years and older than in the group aged 65 to 74 years is also striking. As efforts to improve the process for decision making regarding resuscitation in older patients, the Ministry of Health, Labor and Welfare (MHLW) developed guidelines on the decision-making process pertaining to the last stage of life in 2007 [17]. In 2012, the Japan Geriatrics Society revised their position statement about terminal medicine and care for elderly individuals that had been published in 2001, promoting the use of advanced directives [30]. The MHLW also launched an education program related to engaging in end-of-life discussions for healthcare professionals in 2014 [31]. According to opinion surveys targeting the Japanese public and medical workers in 2017 [32], awareness of advanced directives have increased for these 10 years. The decreasing trends in LSTs shown in our study might be attributed to these efforts. ICU admissions did not significantly decrease. The increase in the number of ICU bed from 2012 to 2014 in Japan [33] might lead to the increases in the demand for ICU admissions.

The decline in administration of LSTs in Japan was inconsistent with the increase in the total cost and the per capita cost of medical care for older population from 2012 to 2014 [34]. The impact of LST utilization on the total cost and the per capita cost of medical care for the population aged ≥65 years might be limited because there is a discrepancy between the study population and evaluation periods for total cost and those used in our study (in the last 7 days for decedents). Expensive or highly advanced medical treatment, such as anticancer agents and long-term hospitalization [9], for older patients may be significantly attributed to the increase in total cost.

Although individual treatment preferences vary, the decline in LSTs just before death suggests a shift in patient- and family-centered end-of-life care. According to a systematic review of family satisfaction with end-of-life care, the absence of CPR before death is associated with a higher rating of the quality of death and dying by family [35]. Opinion surveys targeting the Japanese public and medical workers in 2017 indicated that the preference for withholding LSTs when there is no chance of improvement increased over a 10-year period [32]. A nationwide survey of bereaved families of cancer patients in Japan indicated that the presence of end-of-life discussions to plan care and pain-free or distress-free care in the last days of life are important determinants for quality of care [36]. These studies suggested that the decline in end-of-life care intensity suggests increasing patient and family satisfaction with the quality of death

The frequency of the administration of intensive care measures has been used to evaluate the nature and extent of resource utilization and the costs associated with end-of-life care [6]. Recent developments in large claims databases enabled us to obtain a picture of medical care practices just before death at the national level. Routinely collected claims data can be utilized at a lower cost to study large populations rather than primary data collections. Continuous evaluations of the trends in care can inform efforts to improve care practices.

The study has some limitations. First, because the date of administration is not recorded in claims issued before fiscal year 2012, we could not identify the administration of LSTs or ICU admissions in the last 7 days in the claims issued before 2012. Thus, we could not examine the practice of end-of-life care before 2012, when major policies leading to the dissemination of ACP were developed. We examined only 3 years, from 2012 to 2014. Long-term observations after 2014 are needed to verify the declining trends in the administration of intensive end-of-life care. Second, as our results relied on data from the health insurance claims database, we could not identify the usage of LSTs and ICU admissions not recorded on the submitted claims, e.g., due to a delay in charging. Furthermore, we could not analyze factors that might be associated with changes in practice, such as the cause of death, severity of disease, prior treatments before October of each year, and medical institution characteristics, due to restrictions on the claims used for billing purposes. We focused on intensive care “just before death” by evaluating care in the last 7 days of life. However, patients who were not yet in the last stage of life and benefitted from intensive care could also be included as study subjects. The shortage of medical resources for care might also contribute to the decline. Thus, we could neither examine the appropriateness of LSTs and ICU admissions nor conclude that our results indicate improvement of quality of end-of-life care.

## 5. Conclusions

We elucidated the changes in intensive care practices “just before death” among older patients through a focus on LSTs and ICU admissions in the last 7 days of life, which have not been sufficiently evaluated at the national level. Based on nationally representative claims data in Japan, declining trends in the administration of LSTs one week before death among older inpatients from 2012 to 2014 are shown. Further evidence for the practice of older patients in the dying phase is required. Since 2014, policies for further promoting palliative care and revisions of medical fees to improve end-of-life care have been implemented. Further study of trends after 2014 and continuous nationwide monitoring in care practice by utilizing administrative claims data can inform efforts to improve end-of-life care.

## Figures and Tables

**Figure 1 ijerph-18-03135-f001:**
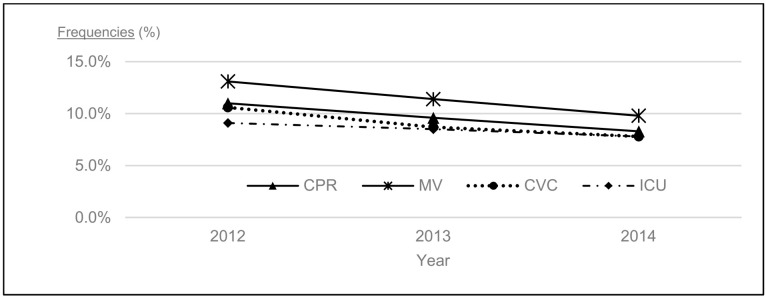
Annual trends in the administration of life-sustaining treatments and intensive care unit admissions in the last 7 days of life from 2012 to 2014. The vertical line represents the frequencies of administration of CPR, MV, CVC, and ICU admissions and the horizontal one year. Abbreviations: CPR: cardiopulmonary resuscitation, MV: mechanical ventilation, CVC: central venous catheter insertion, ICU: intensive care unit.

**Table 1 ijerph-18-03135-t001:** Characteristics of deceased patients in October of 2012–2014.

		2012		2013		2014	
		*N*	%	*N*	%	*N*	%
Total		3362		3473		3516	
Age, years	65–74	632	18.8%	600	17.3%	595	16.9%
	75–84	1244	37.0%	1298	37.4%	1227	34.9%
	85–94	1233	36.7%	1309	37.7%	1391	39.6%
	≥95	253	7.5%	266	7.7%	303	8.6%
Sex	Male	1783	53.0%	1849	53.2%	1876	53.4%
	Female	1579	47.0%	1624	46.8%	1640	46.6%
Cancer	Cancer	1216	36.2%	1326	38.2%	1273	36.2%
Diagnosis	Non-cancer	2146	63.8%	2147	61.8%	2243	63.8%
Ward type	General	2582	76.8%	2634	75.8%	2607	74.1%
	Long-term care	671	20.0%	732	21.1%	782	22.2%
	Psychiatric	93	2.8%	93	2.7%	115	3.3%
	Tuberculosis	12	0.4%	11	0.3%	8	0.2%

**Table 2 ijerph-18-03135-t002:** Annual trends in the administration of life-sustaining treatments and intensive care unit admissions in the last 7 days of life from 2012 to 2014.

		CPR				MV				CVC				ICU			
		**2012**	**2013**	**2014**		**2012**	**2013**	**2014**		**2012**	**2013**	**2014**		**2012**	**2013**	**2014**	
		%	%	%	*p*	%	%	%	*p*	%	%	%	*p*	%	%	%	*p*
Total		11.0	9.6	8.3	***	13.1	11.4	9.8	***	10.6	8.7	7.8	***	9.1	8.5	7.8	0.29
Age																	
	65–74	9.8	9.2	9.7	0.96	16.5	14.2	13.4	0.13	13.6	11.7	8.7	0.01	12.6	11.6	9.1	0.20
	75–84	12.3	11.5	9.3	0.02	15.0	13.6	10.8	**	12.2	9.5	10.4	0.13	9.3	8.2	7.0	0.25
	85–94	10.9	8.6	7.7	**	10.3	9.1	8.6	0.13	8.5	7.5	5.6	**	6.4	7.3	7.9	0.46
	>95	7.9	6.0	4.0	0.05	8.7	5.6	4.3	0.03	4.7	4.5	6.3	0.40	4.0	2.1	5.9	0.63
Sex																	
	Male	12.1	9.6	9.0	**	14.5	12.6	11.2	**	12.2	9.7	8.2	***	10.4	8.4	7.8	0.10
	Fem	9.8	9.5	7.5	0.02	11.4	10.0	8.2	**	8.7	7.6	7.4	0.17	7.1	8.8	7.8	0.72
Cancer	Yes	6.5	4.4	3.7	**	7.0	5.1	4.2	**	12.0	9.0	6.9	***	4.1	2.7	1.8	0.05
Dx	No	13.6	12.8	10.9	0.01	16.5	15.3	12.9	**	9.7	8.6	8.4	0.12	12.8	13.6	12.2	0.75
Wards																	
	Gen	11.2	9.3	8.0	***	14.6	12.6	11.2	***	13.4	10.7	10.2	***				N/A
	LTC	9.5	9.0	7.9	0.28	7.7	7.1	4.9	0.02	0.9	2.0	0.6	0.57				N/A
	Psyc	14.0	19.4	18.3	0.44	9.7	9.7	11.3	0.69	3.2	6.5	3.5	0.99				N/A
	TB	16.7	18.2	0.0	0.32	0.0	18.2	0.0	0.81	0.0	0.0	12.5	0.15				N/A

The value in the column named as “%” represents the frequencies of administration of CPR, MV, CVC, and ICU admissions in each years. *p* represents *p*-value for Cochran–Armitage trend test. The numbers of subjects in the analysis for administration of CPR, MV, and CVC in 2012, 2013, and 2014 were 3362, 3473, and 3516, respectively. The subjects in the analysis for ICU admissions were patients admitted to institutions with acute care or emergency care acute institutions which have ICU. The numbers of subjects in 2012, 2013, and 2014 were 1025, 1018, and 1018, respectively. Symbol: **: *p*-value < 0.01, ***: *p*-value < 0.001. Abbreviations: CPR: cardiopulmonary resuscitation, MV: mechanical ventilation, CVC: central venous catheter insertion, ICU: intensive care unit, Fem: Female, Cancer Dx: Cancer diagnosis, Gen: General ward, LTC: Long-term care ward, Psyc: Psychiatric ward, TB: Tuberculosis ward.

**Table 3 ijerph-18-03135-t003:** Annual decreases in the odds of receiving life-sustaining treatments and intensive care unit admission in the last 7 days of life from 2012 to 2014.

	CPR			MV			CVC			ICU		
	AOR	CI	*p*	AOR	CI	*p*	AOR	CI	*p*	AOR	CI	*p*
2012 (ref)					-				-			
2013	0.88	[0.75–1.03]	0.11	0.89	[0.76–1.03]	0.11	0.82	[0.70–0.97]	0.02	0.99	[0.72–1.35]	0.93
2014	0.74	[0.63–0.87]	***	0.74	[0.64–0.87]	***	0.75	[0.63–0.89]	***	0.88	[0.64–1.21]	0.43

Symbol: ***: *p*-value < 0.001. Abbreviations: AOR: adjusted odds ratio, CI: 95% confidence interval, ref.: reference. *p*: *p*-value for logistic regression analysis using the administration of each LST and ICU admission as a binary categorical response variable and the year of death, age, sex, cancer diagnosis, and ward as categorical variables for adjustment. CPR: cardiopulmonary resuscitation, MV: mechanical ventilation, CVC: central venous catheter insertion, ICU: intensive care unit.

## Data Availability

Restrictions apply to the availability of these data. Data was obtained from the Ministry of Health, Labour and Welfare, and are not available without the permission of the Ministry of Health, Labour and Welfare.

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
