# Peer review of "Decreased Administration of Life-Sustaining Treatment just before Death among Older Inpatients in Japan: A Time-Trend Analysis from 2012 through 2014 Based on a Nationally Representative Sample"

_ijerph, 2021, doi:10.3390/ijerph18063135_

Round 1

Reviewer 1 Report

Thank you for an interesting paper. I have some suggestions and comments that should be adressed in a revised version:

1) Results: Could some of the data be presented using figures? Your manuscript includes many tables with many numbers and are not very easy to read for the readers.

2) Discussion: The discussion is to short. The following themes should be discussed in more detail:

-Inclusion of a broader nternational perspective and comparison with the study results

-Is ACP the reason for the change seen in the results? This should be discussed

-The patients perspective from the literature could be discussed against the results

3) Conclusion:

-The conclusion is very short. I would like to see more about future perspectives of the results. Astatement about the interpretation and the importance of the findings is missing.

4) References: Please check the references. For nr 8 a link is missing

Author Response

Dear Reviewer:

Thank you very much for the thoughtful and constructive feedback regarding our manuscript entitled “Decreased Administration of Life-Sustaining Treatment just before Death Among Older Inpatients in Japan: A Time-Trend Analysis from 2012 through 2014 Based on a Nationally Representative Sample”.

In response to your comments, we made some revisions to the body’s text. The changes to the recently submitted version are indicated by a red font color.

The comments and answers are as follows:

Comment 1
Thank you for an interesting paper. I have some suggestions and comments that should be addressed in a revised version:
1)    Results: Could some of the data be presented using figures? Your manuscript includes many tables with many numbers and are not very easy to read for the readers.

Answer 1:
We presented the annual frequencies of administration of LSTs and ICU admissions in Figure 1 to visualize the temporal trends shown in Table 2. Please see line 204 to 210 in the Results.

Comment 2
2)    Discussion: The discussion is too short. The following themes should be discussed in more detail:
-Inclusion of a broader international perspective and comparison with the study results

Answer 2: 
According to your comment, we make a broader international comparison of the trends in end-of-life care and discuss the background in terms of its consistencies and discrepancies with our results. In the revision, we compare our results with the following findings: 1) there has been a decreasing intensity of care in the United States and European countries at the national level, 2) Japan ranked high in the Quality of Death Index by The Economist Intelligence Unit, and the rank improved in 2015 than in 2010, and 3) other countries had a higher intensity than Japan at the national level. 
For comparison 1), one of the recent leading studies targeting ICU decedents in 14 European countries has been cited for time trend comparison. For comparison 2), we have added comparisons of the Japan’s Quality of Death Index with that of Asian-counties. For comparison 3), we compare our results with those from counties that reported higher intensity than Japan, specifically in the ICU admission and mechanical ventilation in the last 30 days among Medicare decedents and Korean decedents. Although the measurement, survey year, period defined as the end of life, and study settings vary among the references, we hope these comparisons will be useful for interpretation.

The changes are as follow:
Before revision:
“These results are consistent with prior studies reporting decreasing trends in the administration of LSTs and ICU admissions for Medicare recipients who died, in accordance with the development of end-of-life care policies and the popularization and increased availability of advanced directives, hospice care, palliative care [21,22,28]. Our results are also consistent with Japan’s rank in the Quality of Death Index published by The Economist Intelligence Unit, which evaluates developing and developed countries in terms of healthcare spending, the availability of palliative care, public awareness of end-of-life care, the presence of a Do Not Resuscitate (DNR) policy, and so on [7,8]. Japan was ranked 23rd among 40 countries in 2010 [7] and 13th among 80 countries in 2015 [8].”

After revision: (Please see line 249 to 278 in the Discussion.)
“Although the survey year, the period defined as the end of life, and study settings vary among studies, decreasing trends in intensive care are consistent with studies targeting decedents among Medicare recipients and in 14 European countries. Among Medicare recipients 65 or older who died, ICU admissions in the last 30 days of life declined from 2010 to 2015 [22]. The increased penetration of advance care planning and palliative care since 2012 is suggested to have contributed to the decline [22]. Among patients who died in ICUs in 14 European countries, death without withholding LSTs significantly declined from 1999 to 2015 in accordance with the changes in European laws, recommendations and guidelines regarding end-of-life practices over the past decades [28]. Our results are also consistent with Japan’s rank in the Quality of Death Index published by The Economist Intelligence Unit, which evaluates developing and developed countries in terms of palliative and healthcare environment, human resources, affordability of care, quality of care, and community engagement. [7,8]. Japan was ranked 23rd among 40 countries in 2010 [7] and 14th among 80 countries in 2015 [8]. 2 of Asian-counties: Taiwan (at position 6 in 2015) and Singapore (at position 12) which developed a strong and effective national palliative care policy framework ranked higher than in Japan and also improved rank in 2015 than in 2010 [8]. On the other hand, the frequencies of LSTs and ICU admission before the death of patients aged 65 years or older remain lower in Japan than those in countries that reported the care intensity at the national level. ICU admissions in the last 30 days of decedents in 2015 Medicare and decedents [22] in 2009 Korean National Health Insurance Service were 20.0%, and 26.3% [19], respectively, whereas ICU admissions in the last 7 days in the present study were 7.8% in 2014. Administration of MV in the last 30 days of Korean decedents in 2009 was 19.5% [19], whereas MV in the last 7 days in the present study were 9.8% in 2014. Although the period defined as the end of life is longer than our study, differences in healthcare delivery systems may be also related to the discrepancy. Critical care services vary between countries in the number of beds. The number of ICU beds per 100,000 population in 2005 was 20.9 in the USA and 4.3 in Japan [29]. Korean healthcare delivery relies heavily on the private sector because the government allows the private sector to directly respond to the demand for care, leading to an increase in the intensity of care [19].”

Comment 3
3)    Discussion: The discussion is too short. The following themes should be discussed in more detail:
-Is ACP the reason for the change seen in the results? This should be discussed

Answer 3: 
We could not examine whether advance care planning (ACP) was the reason for the change because of the nature of the health insurance claims issued before fiscal year 2012. In Japan, the government developed guidelines on the decision-making process pertaining to the last stage of life in 2007, leading to dissemination of ACP. However, the date of administration is not recorded in claims issued before fiscal year 2012. Thus, we could neither identify the administration of LSTs or ICU admissions in the last days from claims issued before 2012 nor evaluate the change in practice before and after 2007. Thus, we did not draw any conclusions about the reason for change in the Discussion. In the revision, we indicate the data period in which we can identify the administration of LSTs or ICU admissions in the last 7 days, i.e., from 2012 to the present, in the Study design and data source section. We also indicate the absence of claims data for an evaluation of the impact of ACP on end-of-life care as a limitation.

The changes are as follow:
Before revision:
Study design and data source
“As the NDB contains almost 100% of the digitized health insurance claims, its use enables the evaluation of the real-world care practices in nearly the entire older Japanese population [26]. NDB data are untraceable and anonymized. We used the NDB sampling dataset (NDB-SD) and randomly sampled 10% of the inpatient health insurance claims in the NDB for the month of October in each year studied.” 

Discussion
“The study has some limitations. First, we examined care practices over only 3 years, from 2012 to 2014. Long-term observations are needed to verify the declining trends in the administration of intensive end-of-life care. “

After revision:
Study design and data source (Please see line 93 to 96 in the Materials and Methods.)
As the NDB contains almost 100% of the digitized health insurance claims issued after the 2009 fiscal year, its use enables the evaluation of real-world care practices in nearly the entire older Japanese population [26]. NDB data are untraceable and anonymized. The Ministry of Health, Labour and Welfare provides an NDB sampling dataset (NDB-SD) randomly sampled from 10% of the inpatient health insurance claims in the NDB for October, in which seasonal differences in healthcare utilization are not significant for research use. Because the date of administration is not recorded in claims issued before the 2012 fiscal year, we could not identify care practices in the last days of life before 2012. Thus, we used NDB-SD claims data issued from 2012 to 2014.

Discussion (Please see line 343 to 347 in the Discussion.)
This study has some limitations. First, because the date of administration was not recorded in claims issued before fiscal year 2012, we could not identify the administration of LSTs or ICU admissions in the last 7 days in the claims issued before 2012. Thus, we could not examine the practice of end-of-life care before fiscal year 2012, when major policies leading to the dissemination of ACP were developed. We examine only 3 years, from 2012 to 2014. Long-term observations after 2014 are needed to verify the declining trends in the administration of intensive end-of-life care.

Comment 4
4)    Discussion: The discussion is too short. The following themes should be discussed in more detail:
-The patient’s perspective from the literature could be discussed against the results

Answer 4:
We included the interpretation of the decline in intensity from the patient’s perspective referenced in the literature.

After revision: (Please see line 325 to 335 in the Discussion.)
“Although individual treatment preferences vary, the decline in LSTs just before death suggests a shift in patient- and family-centered end-of-life care. According to a systematic review of family satisfaction with end-of-life care, the absence of CPR before death is associated with a higher rating of the quality of death and dying by family [33]. Opinion surveys targeting the Japanese public and medical workers in 2017 [32] indicated that the preference for withholding LSTs when there is no chance of improvement in-creased over a 10-year period. A nationwide survey of bereaved families of cancer patients in Japan indicated that the presence of end-of-life discussions to plan care and pain-free or distress-free care in the last days of life are important determinants for quality of care [34]. These studies suggested that the decline in end-of-life care intensity suggests increasing patient and family satisfaction with the quality of death.”

Comment 5
5)    Conclusion:
-The conclusion is very short. I would like to see more about future perspectives of the results. A statement about the interpretation and the importance of the findings is missing.

Answer 5:
We believe the importance of the findings is their elucidation of changes in intensive care practices “just before death” for older patients through a focus on LSTs and ICU admissions in the last 7 days of life, which have not previously been sufficiently evaluated at the national level.
We think that further evidence for practices for older patients in the dying phase is needed. The identification of temporal trends after 2014 is also needed since policies to further promote palliative care and revisions of medical fees to improve end-of-life care have also been implemented since 2014. Continuous nationwide monitoring of care practice by utilizing administrative claims data is also a future challenge to inform efforts to improve care. We hope that our research will be of interest in other countries developing large administrative claims databases and seeking ways to utilize them to evaluate the actual practice of end-of-life care.
The results can be interpreted to attribute the decreasing trends in the use of LSTs to efforts by governments, academic societies, and healthcare providers to improve end-of-life care. It is also suggested that the decline in LSTs indicates a shift in patient- and family-centered end-of-life care. However, as we examined care practices over only 3 years, from 2012 to 2014, we could not conclude that the reason for the change is past policies. As we could not analyze the appropriateness of LSTs and ICU admissions due to restrictions on the claims used for billing purposes, we also could not conclude that our results indicate the improvement of care.
In the revision, we clearly indicated the importance of our findings and added future perspectives on the results.

Before revision:
“Based on the nationwide claim data in Japan, we identified declining trends in the administration of LSTs one week before death among older inpatients from 2012 to 2014.”

After revision: (Please see line 384 to 393 in the Conclusion.)
“We elucidated the changes in intensive care practices “just before death” among older patients through a focus on LSTs and ICU admissions in the last 7 days of life, which have not been sufficiently evaluated at the national level. Based on nationally representative claim data in Japan, declining trends in the administration of LSTs one week before death among older inpatients from 2012 to 2014 are shown. Further evidence for the practice of older patients in the dying phase is required. Since 2014, policies for further promoting palliative care and revisions of medical fees to improve end-of-life care have been implemented. Further study of trends after 2014 and continuous nationwide monitoring in care practice by utilizing administrative claims data can inform efforts to improve care practices.”

Comment 6
4)References: Please check the references. For nr 8 a link is missing

Answer 6: (Please see line 453 to 459 in the references.)
We added the URL for the 2015 Quality of Death Index Ranking palliative care across the world A report by The Economist Intelligence Unit. 

With these changes to our final manuscript, we hereby resubmit our manuscript for an evaluation. Thank you once again for your feedback.

Sincerely, 
Michi Sakai
Department of Health Informatics, Graduate School of Medicine and Public Health, Kyoto University, Kyoto, Japan, Yoshidakonoe-cho, Sakyo-ku Kyoto, 606-8501, Japan.

Reviewer 2 Report

This is an analysis of routine data representative of Japan with a clearly stated research question. The work is well written and structured, easy to read and conducted in a methodologically sound manner. The authors present as main result a significant decrease of the use of Life Sustaining Treatments (LST) in the last phase of life in people over 65 during the study period. In their discussion, they suggest that this trend may be explained in the context of increased utilization of palliative care related concepts in Japan. The findings are of national and international interest because they touch on an area of health care that is gaining importance worldwide, namely how people of advanced age with non-curable disease should be treated in a medically and ethically sound manner while avoiding over- and undertreatment. I fully recommend a publication of this study and have only have minor suggestions for improvement:

  • Results, first paragraph, line 136: what does "null outcome" mean, how many persons had this outcome?
  • Discussion, last paragraph (line): „Furthermore, we could not analyze factors that might be associated with changes in practice”. This issue should be discussed in more depth. A reduction of LSTs may is basically desirable from a palliative medicine point of view. However, this does not per se mean an improvement in the quality of treatment for the individual patient. Let's take ICU treatment as an example: even very old people can benefit from it, such as previously not or minor impaired people aged 90 plus affected by Covid-19.

Author Response

Dear Reviewer:

Thank you very much for the thoughtful and constructive feedback regarding our manuscript entitled “Decreased Administration of Life-Sustaining Treatment just before Death Among Older Inpatients in Japan: A Time-Trend Analysis from 2012 through 2014 Based on a Nationally Representative Sample”.

In response to your comments, we made some revisions to the body’s text. The changes to the recently submitted version are indicated by a red font color.

The comments and answers are as follows:

This is an analysis of routine data representative of Japan with a clearly stated research question. The work is well written and structured, easy to read and conducted in a methodologically sound manner. The authors present as main result a significant decrease of the use of Life Sustaining Treatments (LST) in the last phase of life in people over 65 during the study period. In their discussion, they suggest that this trend may be explained in the context of increased utilization of palliative care related concepts in Japan. The findings are of national and international interest because they touch on an area of health care that is gaining importance worldwide, namely how people of advanced age with non-curable disease should be treated in a medically and ethically sound manner while avoiding over- and undertreatment. I fully recommend a publication of this study and have only have minor suggestions for improvement:

Comment 1
Results, first paragraph, line 136: what does "null outcome" mean, how many persons had this outcome?

Answer 1:
Hospitalization outcome is recorded as one of the followings “continued”, “cure”, “death”, “termination”, and “transferred” in claims issued in Japan. “Null outcome” means that hospitalization outcome is not recorded in the claims. There were no subjects with no records for hospitalization outcomes. We modified the results of excluded patients.

Before revision
“We excluded patients aged <65 years (n = 2012: 83,380, 2013: 80,956, 2014: 79,423), patients who had an outcome other than death or a null outcome (n = 2012: 136,852, 2013: 139,100, 2014: 142,861), patients with fewer than 7 days of hospitalization (n = 2012: 2,054, 2013: 2,178, 2014: 2,168), and patients with no charge records for treatment (n = 2012: 2, 2013: 2, 2014: 2).”

After revision (Please see line 163 to 164 in the Results.)
“We excluded patients aged <65 years (n = 2012: 83,380, 2013: 80,956, 2014: 79,423), patients who had an outcome other than death (i.e., continued, cure, termination, and transferred) (n = 2012: 136,852, 2013: 139,100, 2014: 142,861), patients with fewer than 7 days of hospitalization (n = 2012: 2,054, 2013: 2,178, 2014: 2,168), and patients with no charge records for treatment (n = 2012: 2, 2013: 2, 2014: 2).”

Comment 2
Discussion, last paragraph (line): Furthermore, we could not analyze factors that might be associated with changes in practice”. This issue should be discussed in more depth. A reduction of LSTs may is basically desirable from a palliative medicine point of view. However, this does not per se mean an improvement in the quality of treatment for the individual patient. Let's take ICU treatment as an example: even very old people can benefit from it, such as previously not or minor impaired people aged 90 plus affected by Covid-19.

Answer 2:
We discussed the limitation in analyzing associated factors of the reduction in LSTs due to restrictions on the claims in more depth.

Before revision:
“Furthermore, we could not analyze factors that might be associated with changes in practice, such as the cause of death, severity of disease, prior treatments before October of each year, and medical institution characteristics, due to restrictions on the claims used for billing purposes.”

After revision: (Please see line 377 to 382 in the Discussion.)
“Furthermore, we could not analyze factors that might be associated with changes in practice, such as the cause of death, severity of disease, prior treatments before October of each year, and medical institution characteristics, due to restrictions on the claims used for billing purposes. We focused on intensive care “just before death” by evaluating end-of-life care in the last 7 days of life. However, patients who were not yet in the last stage of life and benefitted from intensive care could also be included as study subjects. The shortage of medical resources for care might also contribute to the decline. Thus, we could neither examine the appropriateness of LSTs and ICU admissions nor conclude that our results indicate improvement of quality of end-of-life care.”

With these changes to our final manuscript, we hereby resubmit our manuscript for an evaluation. Thank you once again for your feedback.

Sincerely, 
Michi Sakai
Department of Health Informatics, Graduate School of Medicine and Public Health, Kyoto University, Kyoto, Japan, Yoshidakonoe-cho, Sakyo-ku Kyoto, 606-8501, Japan.

Reviewer 3 Report

This paper is concisely and clearly written. The main aspects of the paper that require improvement are as follows:

  1. Methods - this section is a little bit too brief. It would be useful if the authors explained what the Cochran-Armitage test is, how it is performed, and why it is used in this study.
  2. Results - the table formatting has gone awry somewhere with this manuscript. It is important that the tables be fixed so that they are readable - at present, they are almost impossible to read because the text has spilled over various lines. It may be necessary to put them into landscape view in order to fix this.
  3. The other thing that was unclear to me is that the 95% CIs do not appear to be ranges, but single numbers. This requires attention too and the authors should refer to at least some of these CIs explicitly in their discussion of the results so the reader can follow their meaning in the tables.

Author Response

Dear Reviewer:

Thank you very much for the thoughtful and constructive feedback regarding our manuscript entitled “Decreased Administration of Life-Sustaining Treatment just before Death Among Older Inpatients in Japan: A Time-Trend Analysis from 2012 through 2014 Based on a Nationally Representative Sample”.
In response to your comments, we made some revisions to the body’s text. The changes to the recently submitted version are indicated by a red font color.

The comments and answers are as follows:

Comment 1
Methods - this section is a little bit too brief. It would be useful if the authors explained what the Cochran-Armitage test is, how it is performed, and why it is used in this study.

Answer 1:
We applied the Cochran–Armitage trend test as we assume linear trend in the intensive end-of-life care and it is commonly used in epidemiology to test for temporal trend in healthcare utilizations. We used the administration of each LST and ICU admission as binary categorical response variables and the year of death as independent variable. In the revision, we explained what the Cochran-Armitage test is, how it is performed, and why it is used in this study. We also modified the reason for treating the year of death as a categorical variable when conducted logistic regression analysis.  

Before revision:
“We applied the Cochran–Armitage trend test and logistic regression analysis using the administration of each LST and ICU admission as binary categorical response variables and the year of death, age, sex, cancer diagnosis, and type of ward as categorical variables for adjustment. As we did not assume a linear decrease or increase from 2012 to 2014, we treated the year of death as a categorical variable.”

After revision: (Please see line 128 to 139 in the Materials and Methods.)
“We applied the Cochran–Armitage trend test for detecting a temporal trend in the frequencies of LSTs and ICU admissions as we assume linear trend. The Cochran-Armitage (CA) test is used to test whether there is a linear trend when the response is binary. The null hypothesis is that the binomial proportion is the same for all levels of the explanatory variable. It is commonly used in epidemiology to test for temporal trend in healthcare utilizations. We used the administration of each LST and ICU admission as binary categorical response variables and the year of death as explanatory variable. We also conducted logistic regression analysis using the administration of each LST and ICU admission as response variables and the year of death, age, sex, cancer diagnosis, and type of ward as categorical variables for adjustment. As we assume that liner trend is different from 2012 to 2013 and 2013 to 2014, we treated the year of death as a categorical variable using 2012 as the reference year.”

Comment 2
Results - the table formatting has gone awry somewhere with this manuscript. It is important that the tables be fixed so that they are readable - at present, they are almost impossible to read because the text has spilled over various lines. It may be necessary to put them into landscape view in order to fix this.

Answer 2:
We modified the table 2 format. We put all text and values into landscape view.

Comment 3
The other thing that was unclear to me is that the 95% CIs do not appear to be ranges, but single numbers. This requires attention too and the authors should refer to at least some of these CIs explicitly in their discussion of the results so the reader can follow their meaning in the tables.

Answer 3: 
We modified the table 3 format so that the value will appear to be 95% confidence interval.
We also added adjusted odds ratio (AOR) for the decline in LSTs in 2014 relative to 2012 to the discussion.

Before revision: 
“We identified annual declining trends between 2012 and 2014 in the frequencies of the administration of CPR (11.0% to 8.3%), the use of MV (13.1% to 9.8%), and CVC insertion (10.6% to 7.8%) just before death among patients aged ≥ 65 years using nationally representative claims data.”

After revision: (Please see line 242 to 245 in the Discussion.)
We identified annual declining trends between 2012 and 2014 in the frequencies of the administration of CPR (11.0% to 8.3%, adjusted odds ratio in 2014 relative to 2012 (AOR) 0.74 [0.63 - 0.87]), the use of MV (13.1% to 9.8%, AOR 0.74 [0.64 - 0.87]), and CVC insertion (10.6% to 7.8%, AOR 0.75 [0.63 - 0.89]) just before death among patients aged ≥ 65 years using nationally representative claims data.

With these changes to our final manuscript, we hereby resubmit our manuscript for an evaluation. Thank you once again for your feedback.

Sincerely, 
Michi Sakai
Department of Health Informatics, Graduate School of Medicine and Public Health, Kyoto University, Kyoto, Japan, Yoshidakonoe-cho, Sakyo-ku Kyoto, 606-8501, Japan.

Reviewer 4 Report

In this article, the authors describe the temporal trends in the administration of LSTs and ICU admissions just before death in older inpatients in Japan using NDB. The research perspective is important and the method of analysis seems to be appropriate. My comments are mainly about the interpretation of the results:

1. From the results, the authors identified annual declining trends in the frequencies of three LSTs indicator. However, in Japan as a whole, including 2012-2014, both the total cost and the per capita cost of medical care for the elderly continue to increase. From this perspective, please discuss what are the implications of the results of this study.

2. Pease discuss the result that only ICU admissions did not decreased.

3. In the methods section, please explain rationally why only the October data from NDB was used in the analysis.

Author Response

Dear Reviewer:

Thank you very much for the thoughtful and constructive feedback regarding our manuscript entitled “Decreased Administration of Life-Sustaining Treatment just before Death Among Older Inpatients in Japan: A Time-Trend Analysis from 2012 through 2014 Based on a Nationally Representative Sample”.

In response to your comments, we made some revisions to the body’s text. The changes to the recently submitted version are indicated by a red font color.

The comments and answers are as follows:

In this article, the authors describe the temporal trends in the administration of LSTs and ICU admissions just before death in older inpatients in Japan using NDB. The research perspective is important and the method of analysis seems to be appropriate. My comments are mainly about the interpretation of the results:

Comment 1
1. From the results, the authors identified annual declining trends in the frequencies of three LSTs indicator. However, in Japan as a whole, including 2012-2014, both the total cost and the per capita cost of medical care for the elderly continue to increase. From this perspective, please discuss what are the implications of the results of this study.

Answer 1:
We did not focus on the cost of intensive end-of-life care, but we think the moderation of healthcare costs is an important issue for sustainable healthcare systems. We think the impact of LST utilization on the total cost and the per capita cost of medical care for the population aged ≥ 65 years might be limited because there is a discrepancy between the study population and evaluation periods for total cost and those used our study (in the last 7 days for decedents). Expensive or highly advanced medical treatment, such as anticancer agents and long-term hospitalization, for older patients may be significantly attributed to the increase in total cost.
In the revision, we referred to the interpretations of the inconsistency in the decrease in LSTs and the increase in the total cost and the per capita cost for elderly individuals.

After revision: (Please see line 317 to 324 in the Discussion.)
“The decline in administration of LSTs was inconsistent with the increase in the total cost and the per capita cost of medical care for older population from 2012 to 2014 [33]. The impact of LST utilization on the total cost and the per capita cost of medical care for the population aged ≥ 65 years might be limited because there is a discrepancy between the study population and evaluation periods for total cost and those used our study (in the last 7 days for decedents). Expensive or highly advanced medical treatment, such as anticancer agents and long-term hospitalization [9], for older patients may be significantly attributed to the increase in total cost.”

Comment 2
2. Pease discuss the result that only ICU admissions did not decrease.

Answer 2:
In Japan, the number of ICU bed increased from 2012 to 2014. We think it might lead to the increases in the demand for ICU admissions. In the revision, we discussed the possible reason for why only ICU admissions did not significantly decrease.

After revision: (Please see line 314 to 316 in the Discussion.)
“ICU admissions did not significantly decrease. The increase in the number of ICU bed from 2012 to 2014 in Japan [33] might lead to the increases in the demand for ICU ad-missions. “ 

Comment 3
3. In the methods section, please explain rationally why only the October data from NDB was used in the analysis.

Answer:
We used October data because NDB sampling dataset (NDB-SD) only includes claims issued in October in each year. The Ministry of Health, Labour and Welfare provided the data for October in which seasonal differences in healthcare utilization are not significant for research use. In the revision, we modified the description of NDB sampling dataset (NDB-SD) to help readers understand why we used only the October data. 

Before revision:
Study design and data source
“We used the NDB sampling dataset (NDB-SD) and randomly sampled 10% of the inpatient health insurance claims in the NDB for the month of October in each year studied.”

After revision: (Please see line 90 to 93 in the Methods)
Study design and data source
“The Ministry of Health, Labour and Welfare provides an NDB sampling dataset (NDB-SD) randomly sampled from 10% of the inpatient health insurance claims in the NDB for October, in which seasonal differences in healthcare utilization are not significant for research use. “

With these changes to our final manuscript, we hereby resubmit our manuscript for an evaluation. Thank you once again for your feedback.

Sincerely, 
Michi Sakai
Department of Health Informatics, Graduate School of Medicine and Public Health, Kyoto University, Kyoto, Japan, Yoshidakonoe-cho, Sakyo-ku Kyoto, 606-8501, Japan.
